# Sympathetic Innervation Modulates Mucosal Immune Homeostasis and Epithelial Host Defense

**DOI:** 10.3390/cells11162606

**Published:** 2022-08-21

**Authors:** Shilpashree Mallesh, Anne S. Ten Hove, Reiner Schneider, Bianca Schneiker, Patrik Efferz, Jörg C. Kalff, Wouter J. de Jonge, Sven Wehner

**Affiliations:** 1Department of Surgery, University Hospital of Bonn, 53105 Bonn, Germany; 2Tytgat Institute for Liver and Intestinal Research, Amsterdam Gastroenterology Endocrinology and Metabolism, Amsterdam University Medical Centers, Location AMC, University of Amsterdam, 1105 BK Amsterdam, The Netherlands

**Keywords:** sympathetic denervation, epithelial cell function, intestinal barrier function, tight junction, antimicrobial peptides, microbial composition

## Abstract

Intestinal mucosal cells, such as resident macrophages and epithelial cells, express adrenergic receptors and are receptive to norepinephrine, the primary neurotransmitter of the sympathetic nervous system (SNS). It has been suggested that the SNS affects intestinal immune activity in conditions, such as inflammatory bowel disease; however, the underlying mechanisms remain ambiguous. Here, we investigated the effect of SNS on mucosal immune and epithelial cell functions. We employed 6-OHDA-induced sympathetic denervation (cSTX) to characterize muscularis-free mucosal transcriptomes by RNA-seq and qPCR, and quantified mucosal immune cells by flow cytometry. The role of norepinephrine and cytokines on epithelial functions was studied using small intestinal organoids. cSTX increased the presence of activated CD68^+^CD86^+^ macrophages and monocytes in the mucosa. In addition, through transcriptional profiling, the proinflammatory cytokines *IL-1β*, *TNF-α*, and *IFN-γ* were induced, while *Arg-1* and *CD163* expression was reduced. Further, cSTX increased intestinal permeability in vivo and induced genes involved in barrier integrity and antimicrobial defense. In intestinal organoids, similar alterations were observed after treatment with proinflammatory cytokines, but not norepinephrine. We conclude that a loss in sympathetic input induces a proinflammatory mucosal state, leading to reduced epithelial barrier functioning and enhanced antimicrobial defense. This implies that the SNS might be required to maintain intestinal immune functions during homeostasis.

## 1. Introduction

The intestinal epithelium is a single layer of cells, including enterocytes and specialized secretory epithelial cells (IECs), such as Paneth cells [1], goblet cells [2], and enteroendocrine cells [3]. Stem cells residing in the crypt are critical for the rapid self-renewal of this layer [4], which forms a protective barrier against invading antigens, thereby contributing to intestinal immune homeostasis [5]. The epithelial barrier comprises a chemical and physical layer [5]. The chemical barrier is composed of antimicrobial peptides (AMPs), such as cathelicidins, defensins [6], and lectins, such as regenerating islet-derived protein (Reg-)3. *Reg3g* is mainly secreted by Paneth cells and provides spatial separation between the microbiome and the epithelium [7], while defensins protect the intestine against bacterial infection by destroying bacteria or inhibiting bacterial growth [8]. Epithelial cell junctions, the glycocalyx on the microvilli, and mucus produced by goblet cells form the physical barrier inhibiting the invasion of the mucosa by luminal agents [5]. Epithelial barrier integrity is critical, as its dysfunction can lead to inflammation and the pathogenesis of immune disorders, such as inflammatory bowel disease.

Neuronal innervation is known to play a role in maintaining barrier function. The intestine is innervated by the intrinsic neurons of the enteric nervous system and by the axons of the extrinsic sympathetic and parasympathetic nervous systems [9]. While parasympathetic neurons synapse with enteric neurons in the submucosal and myenteric plexuses, it is suggested that sympathetic neurons directly innervate mucosal cells, Peyer’s patches [10], and secondary and ectopic lymphoid structures in the mucosa [11]. This is endorsed by the expression of adrenergic receptors (ARs) receptive to the pre- and postganglionic sympathetic neurotransmitter norepinephrine in various intestinal cell types, including epithelial and mucosal immune cells, such as innate lymphoid cells and tissue-resident muscularis macrophages [12,13]. For years, the SNS has been acknowledged as a distinct regulator of intestinal immunity, as was demonstrated by our group [14] and others [15]. Furthermore, sympathetic innervation was essential to control experimental colitis [14]. A role for the β 2-adrenergic mediate polarization of intestinal muscularis macrophages has been shown upon luminal infection with Salmonella Typhimurium [13]. More recently, our groups showed that muscularis macrophages’ polarization also changes upon surgical intestinal manipulation in an SNS-dependent manner [16].

In contrast to the distinct effect on resident macrophages of the muscularis externa, less is known about the role of the SNS on mucosal homeostasis, although many mucosal cells are known to express ARs. Mucosal macrophages express *β2-ARs* [13], and almost all enterocytes, including specialized epithelial cell subtypes, express a variety of ARs [17,18].

Herein, we aimed to assess the molecular and functional role of the SNS in mucosal homeostasis by the use of a chemical sympathectomy (cSTX) approach based on the application of 6-hydroxydopamine (6-OHDA), an effective model of intestinal muscularis SNS denervation, was recently shown to be superior to both genetic ablations of AR activation and surgical STX [16]. Our data show that cSTX led to a mucosal proinflammatory genotype controlling host defense, indicating a major role of SNS in modulating mucosal homeostasis.

## 2. Materials and Methods

### 2.1. Animals

Eight- to ten-week-old C57BL6/J male mice were used for chemical denervation and the loop model to assess intestinal permeability. The ethics committee of the University of Bonn approved the animal experiments under animal proposal number AZ 81-02.04.2018.A221, 2 April 2018. The mice were maintained under pathogen-free conditions with a 12 h dark/light illumination cycle, temperature of 20–25 °C, and humidity of 45–65%.

### 2.2. Chemical Denervation Procedure (cSTX)

An amount of 80 mg/kg 6-hydroxydopamine (6-OHDA, Sigma Aldrich, Saint Louis, MO, USA), a neurotoxin that targets catecholaminergic nerve terminals [19], was dissolved in 0.1% L-ascorbic acid-containing sterile saline and injected intraperitoneally for three consecutive days. Two weeks after cSTX, the mice were sacrificed by cervical dislocation, and subsequent analyses were carried out.

### 2.3. Flow Cytometry

The muscularis-free mucosal tissue from the small bowel was enzymatically digested for 40 min in a shaking water bath at 37 °C. The enzymatic mixture contained 0.1% collagenase type II (Worthington Biochemical Corporation, Lakewood, NJ, USA), 0.1 mg/mL deoxyribonuclease I (Worthington Biochemical corporation, Lakewood, NJ, USA), 2.4 mg/mL dispase II (Neutral protease grade II, La Roche, Mannheim, Germany), 0.7 mg/mL trypsin inhibitor (PanReac Applichem ITW reagents, Darmstadt, Germany), and 1 mg/mL bovine serum albumin (PanReac Applichem ITW reagents) diluted in sterile, cold PBS. After enzymatic digestion, the cell suspension was filtered with 70 µm gauze, and single cells were obtained. These cells were stained with anti-mouse CD16/32 (Trustain fcX 0.5 mg/mL, Biolegend, San Diego, CA, USA) and fluorochrome-labeled monoclonal antibodies against Alexa fluor 647 anti-mouse Ly6G (0.5 mg/mL, Biolegend), PE anti-mouse F4/80 (0.2 mg/mL, Biolegend), FITC anti-mouse CD45 (0.5 mg/mL, Biolegend), and PE Cy7 anti-mouse Ly6C (0.2 mg/mL, Biolegend) for 30 min at 4 °C. The dead cells were excluded using Hoechst-33342+ (Sigma) dye. Flow cytometry analysis was performed on a Canto II device (FACS Canto, BD Biosystems, Heidelberg, Germany) using FACS Diva software (BD Biosciences, Heidelberg, Germany). The gating strategy is described in the figure legends, and the analysis was performed using the FlowJo software (7.6.5, Tree Star Inc., Ashland, TN, USA).

### 2.4. RNA Sequencing

The total RNA from the vehicle and cSTX muscularis-free mucosal tissue was isolated using a TRIzol-based protocol and was subjected to bulk 3’mRNA sequencing. The sample purity was assessed using a nanodrop, and the ratios for all samples were 1.8–2.1 for A260/A280 and 2.0–2.2 for the A260/A230. The samples were sequenced on an Illumina Hiseq 2500 at the Next Generation Sequencing Facility, University Hospital of Bonn (San Diego, CA, USA). The sequencing data were analyzed using the lexogen pipeline in the Partek software (7.20, Partek, St. Louis, MO, USA), and approximately 13 million reads were generated. The sequences were aligned with STAR 2.5.3a and quantified for mm10- ensemble transcripts release.

### 2.5. 16S Bacterial Sequencing

The stool DNA from mouse ceca was isolated using Qiagen Power fecal Pro DNA QIAamp. The 16S bacterial sequencing of the V3 and V4 regions from the extracted fecal DNA was carried out at Atlas GmBH, Berlin, Germany. Illumina’s BaseSpace App was used for standard analyses to generate reads. In addition, Atlas GmBH was used for bioinformatic analysis to obtain taxonomic bar and diversity plots. The graphs were plotted using GraphPad Prism software version 8.4.3 (Dotmatics, San Diego, CA, USA).

### 2.6. Hematoxylin and Eosin Staining

Freshly dissected intestinal tissues from vehicle- and cSTX-treated mice were fixed in 4% Histofix (Carl Roth, Karlsruhe, Germany) at room temperature. After 24 h, the tissues were rinsed with running tap water for 1 h. The tissues were then embedded in a paraffin block after clearing through Xylene for 1 h. Next, sections were obtained on a microtome at 8 µm thickness and floated in a water bath at 40 °C. Sections were then transferred onto glass slides and stained with hematoxylin and eosin (HE). A Nikon Eclipse TE2000-E microscope was used to capture images at 4× magnification, and histopathological changes were observed.

### 2.7. Immunofluorescence Staining

Paraffin-embedded tissue sections were used to perform immunofluorescence staining. First, the sections were incubated in Triton X (Sigma, Munich, Germany) at room temperature for 15 min and then blocked with 3% BSA (Applicher, Darmstadt, Germany) for 1 h. The sections were then incubated with primary antibodies at 4 °C overnight (TH; Millipore, Burlington, VT, USA), Iba1, and Muc2. Next, the sections were washed 3 times and incubated with the secondary antibodies for 1 h at room temperature. Finally, the sections were washed, embedded with a coverslip, and stored in the dark. A Nikon Eclipse TE2000-E fluorescence microscope was used to capture images at 20× magnification.

### 2.8. RNA Isolation and Quantitative RT-PCR from Mucosal Samples

A TRIzol-based approach was used to isolate the total RNA from the muscularis-free mucosal tissue of vehicle- and cSTX-treated mice. An amount of 1 mL of TRIzol (Life Technologies, Darmstadt, Germany) was added to 20 mg of muscularis-free mucosal tissue and homogenized with ceramic beads. The tissue suspension was incubated with cold chloroform followed by 15 min of centrifugation at max speed. The upper transparent layer was transferred to new tubes and incubated with 0.5 mL of cold isopropanol on ice. The pellet was washed twice with 1 mL of 70% ethanol following centrifugation. The samples were vacuum-dried, and 30 µL of RNAse-free water was added to the pellet and incubated on a thermocycler at 37 °C for 5 min at 550 rpm. cDNA was isolated using a High-Capacity cDNA Reverse Transcription Kit (Life Technologies, Darmstadt, Germany), and quantitative PCR was performed using SYBR green and a real-time PCR detection system.

### 2.9. In Vivo Murine Ileal Loop Model

This in vivo model was performed in BL6 mice for the ileum, as mentioned previously [20]. In short, mice were anesthetized by the inhalation of isoflurane (3–5% at 3–5 L/min flow, AbbVie, Wiesbaden, Germany), and a midline incision of approximately 1 cm was made to open the skin and peritoneum. A segment of the terminal ileum, about 4 cm long, was ligated without disrupting the blood vessels. The stool content was carefully flushed out using ice-cold PBS, and the cut ends were re-ligated using nonabsorbable silk suture 5.0. Next, 1 mg/kg 4 kDa FITC-dextran (70 kDa; Sigma-Aldrich, Munich, Germany) was injected into this loop. The loop was re-inserted into the abdominal cavity, and the peritoneum and skin were closed with two layers of continuous sutures. The mice were anesthetized for 2 h by the inhalation of 1% isoflurane. After 2 h, the fluorescence intensity in the blood serum, which was obtained from cardiac puncture, was measured to check for the presence of FITC-dextran.

### 2.10. Organoid Culture

Small intestinal organoids were obtained from female and male wild-type C57BL/6 mice. First, the small intestines were opened longitudinally, and villi were scraped with a coverslip. Afterward, the intestines were cut into small pieces (2–4 mm), transferred to a 50 mL tube, and washed with 10 mL of cold PBS by pipetting up and down approximately 10 times. Next, the pieces were incubated in 25 mL of 2 mM EDTA in PBS for 30 min at 4 °C while constantly rotating (100 rpm). The supernatant was then removed, and 10 mL of ice-cold PBS containing 10% fetal bovine serum was added and pipetted up and down a few times. The first fraction of the supernatant was discarded, while the 4 subsequent fractions were collected in a 50 mL tube after passing through a 70 μM cell strainer. Next, the crypts were pelleted with a centrifuge for 5 min at 800 rpm and collected in Matrigel (Corning Life Sciences B.V., Amsterdam, The Netherlands) for seeding (20 μL per well; 48-wells plate). The plate was left in an incubator at 37 °C with 5% CO_2_ for 15 to 30 min. Then, 250 μL of complete growth medium was added to the well. Typically, the medium was refreshed every 2–3 days, and organoids were passaged weekly in a split ratio of 1:3 to 1:5, depending on density [21,22].

### 2.11. Microarray Analysis

The small intestinal organoids of mice (*n* = 6) were stimulated for 24 h with the following cytokines: TNF-α at 10 ng/mL (Ref no 315-01A), IL-1β at 10 ng/mL (Ref no 211-11B), and IFN-γ at 10 ng/mL (Ref no 315-05, all PeproTech EC Ltd., London, UK). RNA was extracted from two pooled wells using 500 μL of QIAzol Lysis Reagent (Ref no 79306, Qiagen, Hilden, Germany). After that, RNA was processed for hybridization to Affymetrix Mouse Genome 430 2.0 Array GeneChips (Affymetrix, Santa Clara, CA, USA). Normalization was conducted using MAS 5.0. The analysis of microarray experiments was performed using RStudio software (v1.4, Rstudio, Boston, MA, USA). Comparative analysis was conducted using the limma R package (v3.48.3, Rstudio, Boston, MA, USA).

### 2.12. Quantitative RT-PCR from Organoids

According to the manufacturer’s protocol, the total RNA was isolated from three pooled wells for organoids using Bioline ISOLATE II RNA Mini Kit (GC Biotech, Alphen aan den Rijn, The Netherlands). cDNA was synthesized by using deoxynucleotide triphosphates (Thermo Fisher Scientific, Waltham, MA, USA), random primers (Promega, Leiden, The Netherlands), Oligo dT primers (Sigma-Aldrich Chemie NV, Zwijndrecht, The Netherlands), Revertaid, and Ribolock (both Thermo Fisher Scientific). Quantitative PCR was performed with SensiFAST SYBR No-ROX (GC Biotech) and a CFX Connect Real-Time PCR Detection System (Bio-Rad Laboratories B.V., Lunteren, The Netherlands). Analysis was performed with LinRegPCR software. For normalization, reference genes were used that were previously selected with GeNorm software. Primers were obtained from Sigma. Appendix A shows the primer sequences.

### 2.13. Western Blot

Mucosal samples of 6-OHDA- or vehicle-treated mice were lysed in RIPA buffer (Thermo Fisher Scientific, Waltham, MA, USA), centrifuged at maximum speed for 20 min, and prepared with loading buffer (Biorad, Hercules, CA, USA) to load 30 µg of protein. All samples were processed using the Biorad Western Blot systems (any KD SDS-gels, Trans-Blot Turbo System) and incubated with the antibodies mentioned in the appendix (Claudin-3) overnight at +4 °C. Next, the blot was washed 3 times with PBS-T and incubated with HRP-secondary antibodies (Thermo Fisher Scientific, Waltham, MA, USA) for 2 h at room temperature and imaged with a Biorad ChemiDoc Imaging System.

### 2.14. Statistical Analysis

GraphPad Prism version 8.4.3 software was used to perform the statistical analysis (GraphPad, San Diego, CA, USA). An unpaired *t*-test, or one-way or multiple two-way-comparison ANOVA test were performed, and the significance levels were indicated as *p* ≤ 0.05 (*), *p* ≤ 0.01 (**), and *p* ≤ 0.001 (***). Values were expressed as the mean + SEM.

## 3. Results

### 3.1. Sympathetic Denervation Leads to the Induction of Proinflammatory Genes in the Intestinal Mucosa

To investigate 6-OHDA-mediated cSTX, known to effectively eliminate sympathetic projections towards the muscularis externa also deplete sympathetic innervation of the lamina propria and mucosa, we studied tyrosine hydroxylase-positive (TH^+^) cells within different parts of the intestinal tract two weeks after 6-OHDA or vehicle administration. TH is the rate-limiting enzyme of catecholamines and is, therefore, considered a typical marker of sympathetic neurons. The experimental setup is shown in Appendix A. The efficacy of cSTX on the mucosal level was confirmed in paraffin-embedded tissue sections of small bowel, cecum, and colon samples showing a complete loss of TH+ neurons along the intestinal tract in the cSTX-treated mice compared with the vehicle-treated mice (Figure 1A–C). Enlarged figures from Figure 1A–C are shown in Appendix A. As sympathetic neuronal activity can modulate resident macrophage functions [13,23], we aimed to study the effect of cSTX on these cells. Ionized calcium-binding adaptor molecule 1 (*Iba1*), an intracellular marker of fully differentiated macrophages, was present in the lamina propria mucosa along the intestinal tract, independent of innervation status (Figure 1D–F). Flow cytometry analysis of muscularis-free mucosa revealed a three-fold increase in mucosal CD45^+^ cells after cSTX compared with vehicle-treated mice (Figure 1G–K). As part of the CD45^+^ cell population, MHCII^+^ F4/80^+^ resident macrophages exhibited an increase of five-fold or more in the mucosa after cSTX compared with vehicle treatment (Figure 1L,M). Notably, our previous study showed normal immune cell levels in the muscularis externa two weeks after 6-OHDA treatment, indicating a pronounced role of the SNS in mucosal homeostasis

Furthermore, elevated levels of CD68^+^CD86^+^ cells indicated that macrophages acquired an activated phenotype after cSTX (Figure 1N,O). We also observed a distinct increase in Ly6C^+^Ly6G^−^ monocytes, a subset known to play a critical role in the worsening of inflammation [24] (Figure 1P,Q). In line with the elevated activated macrophage numbers, we observed an increase in the gene expression of proinflammatory markers, such as Tumor Necrosis Factor (*TNF-*)*α* (75-fold), interleukin (*IL-*)*1β* (3.5-fold), and interferon (*IFN-*)*γ* (2.5-fold) (Figure 1R), in muscularis-free mucosa tissue. Simultaneously, strongly reduced Arginase-1 (*Arg1*) and *CD163* gene expression levels were shown in cSTX-treated compared with vehicle-treated mice (Figure 1S).

These data imply that sympathetic denervation leads to, increases, and activates tissue-resident macrophages. Notably, the elevated immune cell levels detected fourteen days after cSTX in the lamina propria seem to be a direct consequence of a missing sympathetic denervation, as our previous study showed that the immunological side effects of the 6-OHDA injection are an immediate, but transient, event in the muscularis externa [16].

### 3.2. Sympathetic Neurons Modulate Epithelial Homeostasis

To acquire a more comprehensive overview of altered molecular pathways after cSTX, we next performed bulk 3′ mRNA sequencing of the mucosal samples of mice treated with 6-OHDA or the vehicle. Principal component analysis (PCA) showed that the individual groups clustered at different axes, indicating significant differences in gene expression (Figure 2A). Functional enrichment analysis of differentially expressed genes (*p* ≤ 0.05 and FDR fold change ± 2) showed enrichment in genes connected to multiple immune-related and host defense gene ontology (GO) terms (Figure 2B). As it has been demonstrated before that sympathectomy induces the molecular signatures of colitis [14], we assessed the gene sets generally associated with immune and epithelial cell functions. Indeed, genes related to the GO-term “epithelial cell development” were enriched (Figure 2C). A heatmap analysis showed differentially expressed genes of the GO term “Epithelial cell development. Additionally, we analyzed the expression of other genes known to be involved in epithelial integrity, antimicrobial function, and barrier integrity by qPCR. Trefoil Factor 3 (*Tff3*), required for epithelial homeostasis by maintaining the surface integrity [25], was increased 16-fold (Figure 2D), and *Ptgs-1* (encoding *COX-1*), critical for keeping cell integrity [26], was increased by 40-fold in cSTX-treated mice when compared with vehicle controls (Figure 2E). Additionally, *Tgfb1* was upregulated significantly (Figure 2F) in tissue after cSTX.

We speculated that, along with the transcriptional changes of genes involved in epithelial development, the tight junction genes and barrier integrity might also be disturbed. Indeed, a heatmap analysis showed transcriptional differences in genes related to tight junctions in cSTX-treated mice (Appendix A). qPCR analysis showed that claudin and occludin expression in mucosal samples of cSTX-treated mice were increased (*Cldn3* (seven-fold), *Cldn2* (three-fold), and *Ocln* (two-fold)) (Figure 2G,J,K). In line with this, we also observed an increase of more than five-fold in claudin-3 protein expression in cSTX-treated mice (Figure 2H,I), confirming altered transcriptional regulation of intestinal barrier-maintaining genes. We next assessed functional intestinal integrity in vivo by permeability measurement. First, a loop of the murine terminal ileum was ligated without disrupting the blood supply. Afterward, fluorescein isothiocyanate (FITC)-dextran was injected into the isolated loop (Figure 2L). After 2 h, the mice were sacrificed, and the fluorescence levels in the plasma were measured and showed to be increased in mice subjected to cSTX compared with vehicle-treated controls (Figure 2M). These findings verify disturbed barrier integrity after cSTX.

Notably, although these data show evident inflammatory induction, the effects are limited to a molecular level and do not induce morphological changes, since hematoxylin and eosin stainings showed no visible signs of pronounced inflammation or tissue damage in cSTX-treated versus vehicle-treated mice (Figure 2N).

### 3.3. Sympathetic Denervation Induces Expression of Antimicrobial Defense Genes

The enhanced expression of the proinflammatory genes *IL-1β*, *TNF-α*, and *IFN-γ*, and epithelial repair genes, such as *TFF3*, together with the loss of functional barrier integrity associated with cSTX, prompted us to further investigate the role of the sympathetic innervation in the host defense mechanisms of the epithelial cells. As a part of the host defense strategy, the epithelium produces a protective mucus layer and antimicrobial peptides. We studied the expression of *Muc2*, the gene responsible for the mucus-forming protein mucin in the intestinal tract, by qPCR and observed a three-fold increase in the mucosal samples of cSTX-treated mice (Figure 3A). In contrast to the gene expression data, *Muc2* (Appendix A) protein expression was diminished after cSTX. In line with this, alcian blue-positive goblet cells were reduced following denervation (Appendix A). Furthermore, differential expression analysis showed the hierarchical clustering of antimicrobial genes connected to the GO term “antimicrobial humoral response” (Figure 3B). Antimicrobial defense genes, including alpha-defensins (*Defa*) and *Reg3g*, were among the significantly cSTX-enriched genes (Figure 3C). The qPCR data demonstrated increased expression of *Defa1* (6-fold), *Reg3g* (4-fold), *Defb3* (3.5-fold), and *Lyz1* (3.5-fold) in cSTX-treated mice (Figure 3D). We interpret these findings as the potential involvement of the SNS in the modulation of the intestine’s antimicrobial defense.

### 3.4. Sympathetic Neuronal Activity Impacts Epithelial Cell Functions Indirectly via Cytokines

We next investigated whether sympathetic neuronal activity affects epithelial gene expression directly or indirectly through other cell types. To test this, we generated small mouse intestinal organoids and basolaterally stimulated these with norepinephrine. After 72 h of stimulation, we observed no differences in the mRNA levels of *Cldn2, Cldn3*, *Defa1, DefB3, Muc2, Ocln, Reg3g* and *Tff3* when compared with the vehicle-stimulated organoids (Appendix A). The in vivo studies demonstrated increased gene expression of the cytokines *IFN-γ*, *IL-1β* and *TNF-α,* which are likely derived from mucosal cells. We hypothesized that sympathetic neuronal activity affects epithelial gene expression and function via immune-cell-secreted cytokines. To that end, we performed a microarray analysis of organoids stimulated with cytokines that we found to be upregulated after cSTX. Organoids were cultured in equal settings, and twenty-four hours after stimulation, the genes were analyzed for differential expression comparing cytokine-stimulated and vehicle-stimulated organoids. The top 30 up- and downregulated genes are shown in heat maps (Figure 4A). From the analyses of transcriptional profiling, we observed an increase in the expression levels of *Cldn2, Cldn3*, and *Reg3g* after stimulation with *IFN-γ* and *TNF-α* when compared with vehicle-stimulated organoids (Figure 4B). Treatment with *IL-1β* did not affect any of these genes. Further investigation with gene set enrichment analyses (GSEA) (Appendix A) showed the enrichment of genes expressing tight junctions following treatment with *IFN-γ* (normalized enrichment score (NES) = 1.96; *p* < 0.001) and antimicrobial peptides upon treatment with *TNF-α* (NES = 2.04; *p* < 0.001) when compared with non-stimulated organoids (Figure 4C). In addition, treatment with *IL-1β* led to an enrichment in antimicrobial peptide genes compared with non-stimulated organoids (NES = 2.09; *p* < 0.001). These data indicate that specific cytokines can modulate the gene expression patterns of tight junctions and antimicrobial peptides in intestinal epithelial organoids. As the cytokines *IFN-γ* and *TNF-α* mainly originate from immune cells in vivo, our data suggest that the SNS exerts its supportive functions on epithelial barrier function by affecting mucosal immune cell cytokine production.

### 3.5. Sympathetic Denervation Alters the Intestinal Microbial Composition

As we demonstrated, epithelial cell function and AMP production are regulated through the SNS, and, conversely, recent studies showed elevated sympathetic activity after antibiotic treatment [27]; therefore, we tested the effect of cSTX on microbial composition in the intestine. To that end, we analyzed how the microbiome in cSTX-treated mice would be altered through the 16S sequencing of the microbial samples of cecum two weeks after 6-OHDA or vehicle treatment using 16S sequencing. PCA showed that the individual groups clustered separately, indicating an effect of cSTX on the microbiome’s composition (Figure 5A). However, the microbial species richness was similar between the cSTX- and vehicle-treated mice (Figure 5B). Likewise, both groups had an equal median Shannon index value, indicating similar species evenness upon cSTX (Figure 5C). The analysis of lower phylogenetic levels did not show quantitative changes in phylum (data not shown). However, significant changes in cSTX samples occurred at the order level. An increase in Bacilli Erysipelotrichales and a reduction in the Clostridia Vadin BB group species were observed (Figure 5D,F). Further, Bacilli Erysipelotrichaceae was increased, and two species of Clostridia, Ruminococcaceae and Vadin BB group, were reduced at the family level (Figure 5E,G). Similarly, we observed an increase in Bacilli Faecalibaculum and reduction in the Clostridia Vadin BB group at the genus level (Figure 5H). These data suggest that the loss of sympathetic neurons led to increased Bacilli compositions, while the Clostridia species were reduced. Thus, the microbiome composition seems to be connected to a properly functioning intestinal sympathetic neuronal system.

## 4. Discussion

Sympathetic neuronal innervation projects to the intestinal mucosa and profoundly influences intestinal functions. While the interaction between IECs and immune cells in mucosal homeostasis has been studied extensively [28,29,30], the role of SNS in this interplay remains unexplored. In this study, we focused on the immunomodulatory role of the SNS in the mucosa, with a particular interest in IECs. We previously showed that STX could effectively be induced chemically in the intestinal muscularis using the neurotoxin 6-OHDA, which leads to a reduced number of TH^+^ neurons [14,16]. As these studies described an immunomodulatory role of the SNS in regulating muscularis macrophage functions in the muscularis externa, we aimed to investigate the effect of the SNS on mucosal immune homeostasis and epithelial functions. We performed flow cytometry analysis on muscularis-free mucosal tissue two weeks after denervation. An increased number of MHCII^+^ F4/80^+^ and CD68^+^ CD86^+^ cells [31] was found, indicating the role of the SNS in the regulation of mucosal macrophage functions.

Interestingly, we also found increased Ly6C^+^ Ly6G^−^ monocyte numbers in the mucosa two weeks after denervation. This is contrary to our previous observations in the muscularis layer, where we observed an increase in monocyte numbers on day four after denervation that rapidly dropped to the control levels [16]. This implies that the role of the SNS in immune cell modulation is tissue-specific and different between the muscularis and mucosal layers. Further, the classical proinflammatory genes (e.g., *IFN-γ*, *IL-1β* and *TNF-α*) were upregulated, and anti-inflammatory genes were downregulated (e.g., *Arg1* and CD163) [32] in the mucosa after denervation. Although the cellular source of the cytokines and inflammatory markers has not been studied, we speculate that it is the resident cell immune cells, particularly macrophages, that are abundantly present (Figure 1D–F) along the complete intestine’s lamina propria mucosae.

IECs contribute to the developing immune system by producing cytokines and chemokines in response to commensal bacteria [29,32]. The mucosal macrophages located beneath the epithelial layer initiate the adaptive immune system and play a role in sampling and clearing pathogenic bacteria [33]. Furthermore, IECs secrete mucins and AMPs to support and maintain the intestinal barrier and homeostasis and provide pathways for the delivery of luminal bacteria to these antigen-presenting cells [28]. As we observed changes in the immune cell numbers upon cSTX, we investigated its impact on the barrier and defensive functions. Indeed, we found multiple lines of evidence that the sympathetic mucosal innervation controls barrier integrity and epithelial antimicrobial responses. Firstly, and most importantly, the epithelial permeability was diminished upon SNS, as shown in a FITC dextran loop model [34]. Secondly, we found an increased expression of *Cldn3*, *Cldn2* and *Ocln* upon denervation. These genes code for a protein involved in tight junction formation, which is necessary to connect adjacent epithelial cells, regulate intestinal permeability, and is crucial for intestinal barrier function [35,36]. Although these findings show that the barrier permeability is affected, it seems controversial that tight junction gene expression and permeability increase simultaneously. As we confirmed tight junction molecule expression at the transcriptional level and with claudin-3 also representative at the protein level, we interpreted the increased permeability in cSTX-treated animals as a consequence of impaired tight junction assembly. The induced gene expression of tight junction genes indicates that compensatory mechanisms are required to counteract the detrimental effects of cSTX on the integrity of the intestinal barrier.

Increased expression of *Muc2* and *Tff3* after cSTX was observed. *Muc2* expression supports the mucous layer at the epithelial surface, and *Tff3* modulates the intestinal physical barrier by offering structural integrity to mucus and promoting IEC migration and repair [37]. The role of *Muc2* seems to be crucial in intestinal inflammatory disorders, as Muc2-deficient mice show epithelial cell structure deformation, leading to increased inflammatory cell infiltration and developing spontaneous colitis [38,39]. Furthermore, these mice showed increased *IL-1β* and *TNF-α* expression levels compared with wild-type controls. In line with this, we observed a reduced number of goblet cells and *Muc2* protein expression following cSTX, highlighting the role of sympathetic neurons in mucosal immune disorders. Acute intestinal wound healing also requires *Muc2*, and *Muc2*-deficient mice have severe healing disturbances of intestinal anastomosis [40], highlighting their role in mucosal homeostasis [41]. The third level of evidence for a protective barrier role of the SNS comes from our observation of increased mRNA levels in AMP genes. Several AMPs belonging to the class of alpha- and beta-defensins and cathelicidins were induced upon cSTX. Alpha-defensins comprise most of the secretory granules of Paneth cells but are also produced by immune cells [42]. While beta-defensins are mainly produced by epithelial cells [43], the high presence of alpha-defensins transcripts and simultaneous induction of *Reg3g* indicates that the AMP response might be derived from the Paneth cell compartment or immune cells. Although our data provide evidence of a possible contribution of the SNS in the regulation of the antimicrobial host defense at the epithelial barrier, final proof of increased host defense mechanisms requires future studies, including bacterial killing capacity assays, which have been previously used to measure the overall antimicrobial capacity of luminal contents [44].

After the in vitro norepinephrine stimulation of mouse organoids, which do not contain immune cells, but include Paneth cells as part of the epithelial cell populations, we did not observe any differences in the tight junction or AMP expression levels compared with the controls. This prompted us to investigate whether SNS indirectly impacts epithelial cells, i.e., via cytokines secreted by immune cells. Indeed, organoid stimulation with the proinflammatory cytokines *IFN-γ*, *IL-1β* and *TNF-α* led to elevated levels of *Cldn2*, *Cldn3* and *Reg3g*, indicating that immune cells might serve as a linkage between sympathetic neurons and epithelial cells through the secretion of inflammatory cytokines. This is in line with previous studies highlighting the cytokine-mediated secretion of AMPs and modulation of alternative epithelial cell functions, as reviewed in [45,46]. It is also explanatory for the expression of cytokine receptors, such as *IFN-γR* [47] and *TNFR1* [48], in epithelial cell types. The finding that enhanced *IFN*-*γ* signaling on epithelial cells is associated with epithelial damage [47] further supports that SNS–mediated inflammation is causative and not the consequence of cSTX-induced mucosal barrier dysfunction.

The change in epithelial biology, particularly the induced antimicrobial host defense, suggested that the sympathetic denervation might also impact the gut microbiome, as a shift in the gut microbiome is likely associated with the pathophysiology of the gut [49]. Although the role of SNS in gut dysbiosis through interaction with epithelial and immune cell populations is not well understood, recent studies have described the connection between the SNS and the microbial composition [23,50], and their impact on mucosal physiology. Herein, we found supportive evidence of a relationship between the SNS and the intestinal microbial composition. Though we did not observe significant changes in the higher hierarchical levels, we observed distinct changes upon denervation at the order, family, and genus levels. Bacilli, Erysipelotrichaceae, Clostridia Ruminococcaceae, and Clostridia Vadin BB were among the microbiota with different levels between cSTX- and vehicle-treated mice.

Interestingly, the more abundant Gram-positive Erysipelotrichaceae are associated with inflammation-related gastrointestinal disorders [51]. For example, they have been shown to increase in the lumen of colorectal cancer patients [52]. In inflammatory bowel disease, the findings seem inconsistent, as some researchers observed an inflammation-associated increase [53,54], while others observed a decrease [55]. While it remains unclear if the observed microbial changes upon cSTX treatment are causative of the altered immune and epithelial functions, our findings show that the SNS can shape microbial composition. Our results suggest that, besides recent findings of an effect of microbiota on sympathetic neuronal functioning [27], a reciprocal interaction of the SNS affecting the luminal microbiome might also be relevant. However, more studies are needed to decipher the molecular interactions and relevance of the SNS–microbiome counteractions.

## 5. Conclusions

In summary, we show that the SNS plays an essential role in modulating mucosal homeostasis. Sympathetic depletion abrogated anti-inflammatory genes in the intestine and shaped the basic mucosal profile to a proinflammatory pattern. In addition, tight junction and antimicrobial defense genes were elevated upon denervation. The link between increased immune cell numbers and the upregulation of genes responsible for epithelial barrier functions and distinct microbial changes suggests an overarching role of the SNS in regulating intestinal immune homeostasis, epithelial function, and microbial composition. Further studies in this field are needed to understand the role of this link in gastrointestinal diseases, including intestinal mucosal healing, inflammatory bowel disease, and colorectal cancer.

## Figures and Tables

**Figure 1 cells-11-02606-f001:**
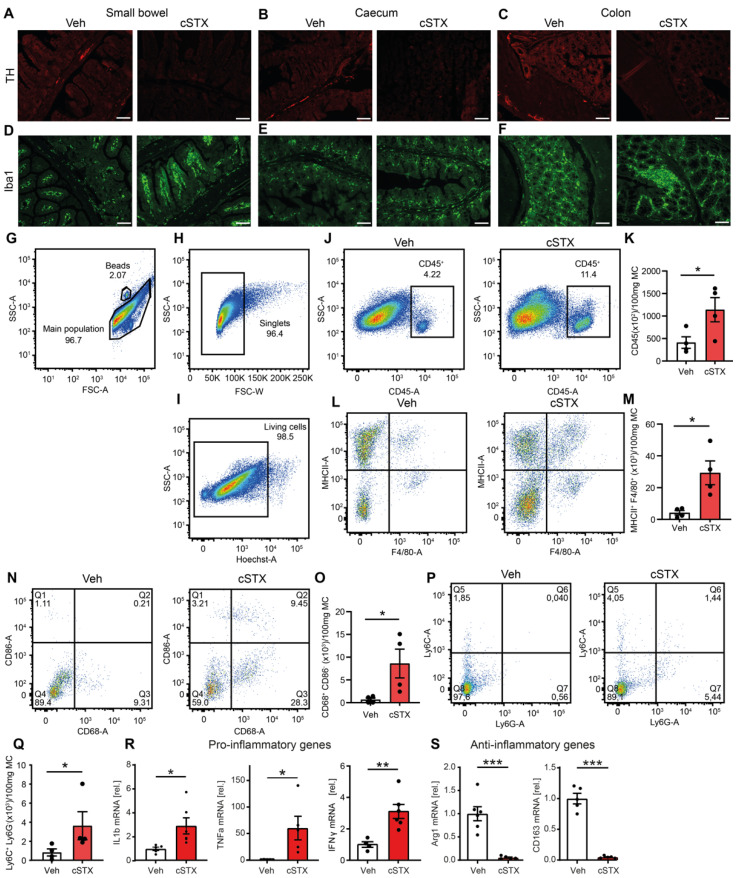
Proinflammatory gene expression in the mucosa after cSTX. Mice were injected with 6-OHDA to achieve a chemical sympathectomy (cSTX) within the intestinal lamina propria. Controls were injected with a vehicle (Veh). (**A**–**C**) Two weeks after the last injection, cSTX efficacy was visualized by immunofluorescence staining for *TH*, showing a complete loss of TH^+^ fibers in the ileum (**A**), cecum (**B**), and colon (**C**) upon cSTX, Scale bars, 100 μm. (**D**–**F**) Immunofluorescence staining for Iba1 shows the presence of Iba1^+^ macrophages throughout the ileum (**D**), cecum (**E**), and colon (**F**) in the Veh- and cSTX-treated mice Scale bars, 100 μm. (**G**–**I**) Representative FACS dot plots of muscularis-free lamina propria preparations of cSTX- and Veh-treated mice, demonstrating the gating strategy based on forward/sideward scatter (**G**), singlets (**H**), and Hoechst live–dead exclusion (**I**). (**J**,**L**,**N**,**P**) Representative FACS dot plots showing CD45^+^ immune cells (**J**), the ratio of MHCII^+^ F4/80^+^ (**L**), the percentage of CD86^+^ vs. CD68^+^ (**N**), and the percentage of Ly6C^+^ vs. Ly6G^+^ cells (P) upon cSTX as compared with the Veh group. (**K**,**M**,**O**,**Q**) Bar graphs showing the quantification of the CD45^+^ cell numbers (**K**) *n* = 4, MHCII^+^ F4/80^+^ cell numbers (**M**) *n* = 4, CD86^+^ C68^+^ cell numbers (**O**) *n* = 4, and Ly6C^+^ Ly6G^−^ monocytes (**Q**) *n* = 4. (**R**,**S**) mRNA levels of proinflammatory genes (**R**) and anti-inflammatory genes (**S**) *n* = 6 measured by qPCR in mucosal tissue samples of Veh- (white) and cSTX-treated mice (red). Values in each column are displayed as the mean ± SEM, and an unpaired *t*-test was carried out for statistical analysis (* *p* < 0.05, ** *p* < 0.01, and *** *p* < 0.001).

**Figure 2 cells-11-02606-f002:**
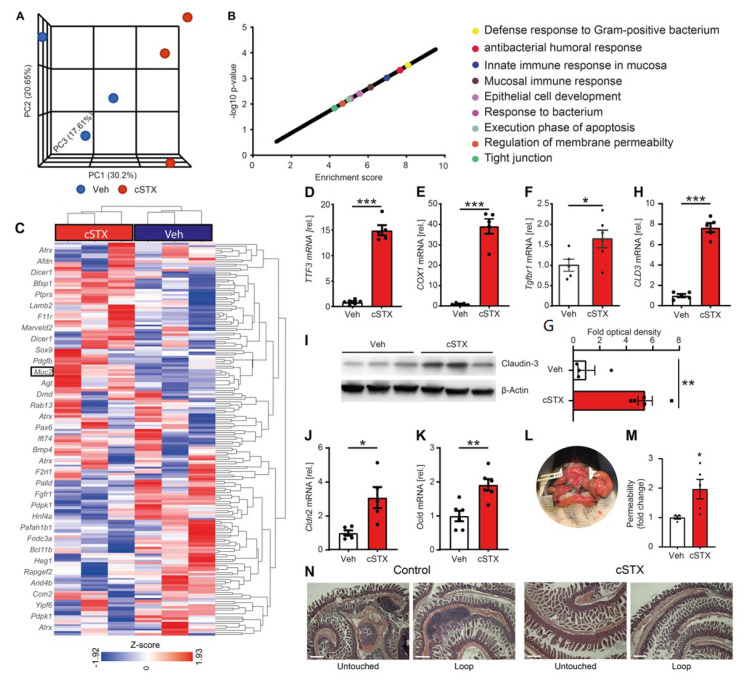
The primary immune status of the intestinal mucosa is altered after cSTX. Mice were injected with 6-OHDA to achieve chemical sympathectomy (cSTX). Controls were injected with the vehicle (Veh). The muscularis-free lamina propria specimen underwent 3’mRNA bulk sequencing two weeks after the last injection. (**A**) PCA plot showing the clustering vehicle and cSTX samples (*n* = 3). (**B**) Functional enrichment analysis of the differentially expressed genes identifying several GO terms related to the host’s immune response and defense mechanisms. (**C**) Heat map showing the fold-changes in genes related to the GO term’s epithelial cell development. (**D**–**F**) The mRNA levels of mucosal repair genes were measured by qPCR in the mucosal samples of vehicle- (white) and cSTX-treated mice (red). (**G**,**J**,**K**) The mRNA levels of tight junction genes were measured by qPCR in the vehicle- (white) and cSTX-treated mice (red) mucosal samples. (**H**) cSTX-treated mice showing an increase in claudin-3 protein expression as compared with vehicle-treated mice. (**I**) Quantification of protein expression normalized to β actin showing an increase in the claudin-3 protein in cSTX-treated mice (*n* = 5) compared with vehicle treatment (*n* = 4). (**L**) Loop model showing a 4 cm ligation at the terminal ileum. (**M**) Increased intestinal permeability in cSTX mice (red) compared with the vehicle-treated mice (white) measured by the fluorescence in the plasma. (**N**) Compared with vehicle-treated mice, hematoxylin and eosin staining of intestinal tissue showed no visible signs of damage upon cSTX treatment. *n* = 5. Scale bars, 200 μm Values in each column are displayed as the mean ± SEM, and an unpaired *t*-test was carried out for statistical analysis (* *p* < 0.05, ** *p* < 0.01, and *** *p* < 0.001).

**Figure 3 cells-11-02606-f003:**
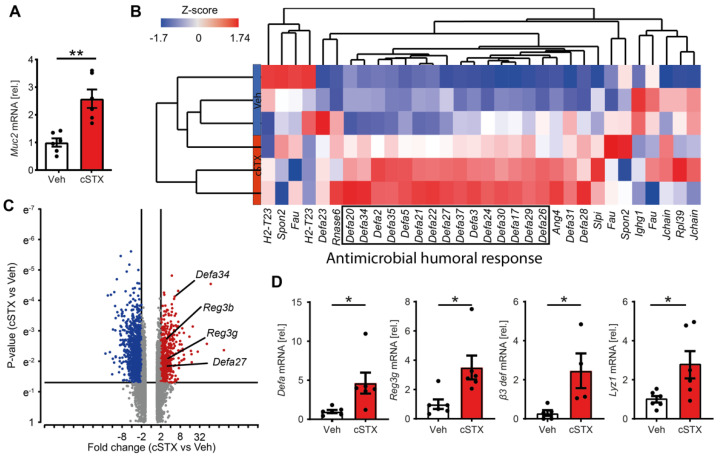
cSTX elevates tight junction and antimicrobial gene expression in the mucosa. Mice were injected with 6-OHDA to achieve a chemical sympathectomy (cSTX) within the intestinal lamina propria. Controls were injected with a vehicle (Veh). The muscularis-free lamina propria specimen underwent 3’mRNA bulk sequencing and qPCR analysis two weeks after the last injection. (**A**) The mRNA levels of the Muc-2 gene were measured by qPCR in the mucosal samples of vehicle- (white) and cSTX-treated mice (red) *n* = 6. (**B**) Heat map displaying the hierarchical clustering of differentially expressed genes related to the antimicrobial humoral response GO term (*n* = 3). (**C**) Volcano plot showing up- and downregulated genes and highlighting antimicrobial defense genes in the cSTX compared with the Veh group. (**D**) Relative mRNA induction (normalized to the Veh group) of individual antimicrobial defense genes was measured by qPCR in the mucosal samples of vehicle- (white) and cSTX-treated mice (red) *n* = 6. (Values in each column are displayed as the mean ± SEM, and an unpaired *t*-test was carried out for statistical analysis (* *p* < 0.05 and ** *p* < 0.01).

**Figure 4 cells-11-02606-f004:**
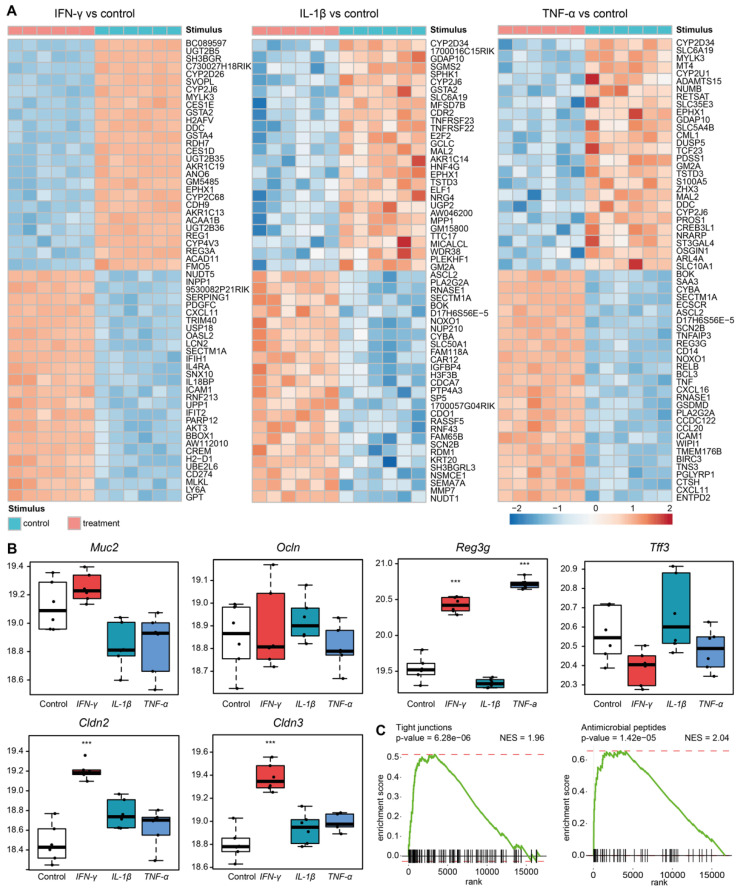
Microarray analysis of mouse small intestinal organoids stimulated with *IFN-γ*, *IL-1β* and *TNF-α*. (**A**) Heat maps showing the top-30 up- and downregulated genes upon stimulation of organoids with cytokines. (**B**) Expression levels (log2) of *Muc2, Ocln, Reg3g, Tff3, Cldn2* and *Cldn3* in organoids after stimulation with vehicle or the cytokines *IFN-γ*, *IL-1β* and *TNF-α* (10 ng/mL). *n* = 6. (**C**) Gene set enrichment analyses (GSEA) for tight junctions after IFN-y treatment (left) and antimicrobial peptides after *TNF-α* treatment (right) with normalized enrichment scores (NES) of 1.96 and 2.04, respectively. The Y-axis shows the enrichment score, while the X-axis represents the specific gene ontology (tight junction or antimicrobial peptide) set genes. The green line connects the points of the enrichment score and genes. Normalization was conducted using MAS 5.0. Analysis of microarray experiments conducted performed using RStudio software (v1.4). Comparative analysis was conducted using the limma R package (v3.46.0). Moderated t statistic was used for statistical analysis.

**Figure 5 cells-11-02606-f005:**
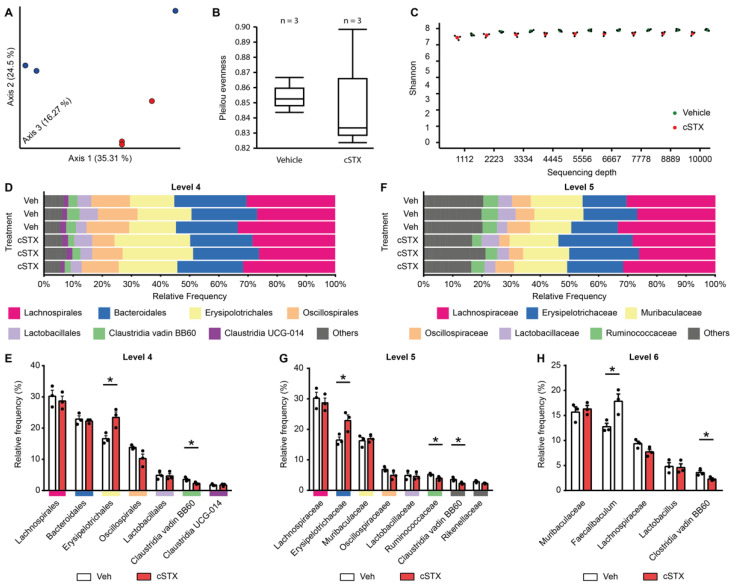
Intestinal sympathetic neurons modulate a specific bacterial population. C57BL/6 mice who underwent saline or 6-OHDA treatment were sacrificed 17 days later (*n* = 3). DNA was isolated from the cecum stool samples and was subjected to 16S rRNA sequencing. (**A**) PCA shows the clustering of individual groups (Veh in blue and cSTX in red). (**B**,**C**) Microbial richness (**B**) and median Shannon’s index (**C**) were similar between Veh and cSTX groups. (**D**,**F**) Stacked row diagram showing the relative frequency of bacterial composition at the levels of order (**D**) and family (**F**) in the vehicle (white) and cSTX-treated mice (red). (**E**,**G**,**H**) Bar graphs with individual data points show significant changes in Bacilli Erysipilotrichales and Clostridia Vadin BB60 composition at the levels of order (**E**), family (**G**), and genus (**H**). Values in each column are displayed as the mean ± SEM, and an unpaired *t*-test was carried out for statistical analysis (* *p* < 0.05).

## Data Availability

The data presented in this study are available on request from the corresponding author.

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
