# Peer review of "Sympathetic Innervation Modulates Mucosal Immune Homeostasis and Epithelial Host Defense"

_cells, 2022, doi:10.3390/cells11162606_

Round 1
Reviewer 1 Report
The field of investigation proposed and studied by he authors are interesting, novel and important, however, several of the conclusions are not well supported by the experimental design or by the figures, which by the way, are of very low quality. Below are my comments and suggestions to improve this manuscript.
- Introduction. Ref 10 refers to sheep ileum and not mouse. Ref 11 and 12, please provide original articles, not reviews.
- Methods. A) Why not just feed FITC-Dextran instead of doing this invasive procedure? B) Organoids. It seems authors did not prepare organoids, but rather single cell suspension culture as it is not described the use of a matrigel or something similar to form the organoids. Authors need to clarify this or their results in which they use "organoids" cannot be accepted.
- Fig. 1A-C. Quality of figures are really bad. It is not clear to me whether the staining of TH+ cells is real. It looks more as an autofluorescence, which is very high in the intestine. Unfortunately, this compromises the whole study once is not possible to determine for sure that cSTX worked. This figure must be improved to clearly demonstrate TH staining.
- Fig. 1D-F. Again, images do not clearly show the staining of Iba1. The autofluorescence is huge and the quality of the images poor.
- Fig. 1 L, M. Since it is not possible, based on the IF images provided in Fig. 1A-C, to confirm that TH+ cells were affected, I'm not sure if the effect on macrophages authors describe in this figure is because of the loss of sympathetic innervation or an unspecific direct effect of the 6-OHDA on mucosal macrophages. Another method of sympathectomy is recommended for comparison.
- Fig. 1P, Q. This may not be the case in the gut. This ref describes these cells in spinal cord injury. The gut is a totally different environment. I suggest authors to remove this from the manuscript or investigate it further.
- Fig. 2J. Loop of the terminal ileum to investigate barrier integrity is a very artificial and invasive technique. I would like to see the classic method for permeability investigation which is simply feed mice with FITC-Dextran and check fluorescence 2 to 4 hours later in the serum.
- Lines 295, 296. Authors state that no inflammation was observed by H&E staining. I don't understand how this is possible with a 75-fold increase in TNF-a for example in STX mice.
- Lines 297-299. How do authors connect these findings with increased macrophages and increased pro-inflammatory cytokines? Are these cytokines coming from macrophages? The purpose of this study is to show neuroimmune-epithelial cell interaction, but this is not clearly demonstrated. Authors should address this by inducing STX in mice deficient for intestinal macrophages. As macrophages are mononuclear phagocytes and express CX3CR1 in the gut, a good model would be depleting intestinal macrophages by using for example CX3CR1-CreERT2+/–iDTR+/– mice.
- Line 361. Top 30 genes rather than top 20 genes are shown.
Reviewer 2 Report
I fully support this study, and I have no objection.
Reviewer 3 Report
The manuscript of Mallesh et al shows the the implication of SNS in intestinal mucosal homeostasis. The manuscript is well written and the study is well developed. The results are clearly presented and are quite relevant to the field. I have only few questions about.
-Why are there two different sections for explaining quantitative PCR (2.8 and 2.12)? Authors should included the type of sample they use for each
-Authors should add the number of samples (n) for each figure
-Authors describe transcriptional differences in genes related to tight juctions, but they do not verify the protein expression. It wuould be recommended to measure protein expression by western blot or stainings.
-In line with the previous point, the authors say "qPCR analysis showed thata claudin and occludin expression in mucosal samples of cSTX-mice were increased", that means to me, an increase of tight junctions and also the intestinal barrier. However, the next experiment indicates an increased in intestinal permeability and the authors conclude "disturbed barrier integrity after cSTX. These findings look contradictory and need a better explanation. Moreover, a protein quantification could clarify this point.
-In the 3.3. section, the authors show the expression of antimicrobial defense genes and based on these results they conclude "these data indicate a role of the SNS in modulating epithelial host defense functions" (lines 333-334). I think is a bit pretentious. In addition, it woul be necessary to mesure those factors in intestinal explants as all these results are based only in mRNA levels.
Round 2
Reviewer 1 Report
Thanks for addressing my concerns.
Please change commas in the facs plots to dots.